# META-EVALUATION COLLAPSE: WHO JUDGES THE JUDGES OF JUDGES?

## ABSTRACT

Large language models (LLMs) are increasingly used as evaluators, yet their reliability as "judges" remains poorly understood. We introduce the concept of *meta-evaluation collapse*: recursive LLM-based evaluation converges toward internally consistent but fragile fixed points that are detached, from human or domain-grounded truth. Through an *operator-theoretic* analysis, we show that unanchored evaluation hierarchies inevitably contract to *biased equilibria*, either collapsing into trivial consensus or amplifying systematic preferences such as fluency over accuracy. Empirically, using multilingual health queries, we find that LLM judges display high inter-model agreement but drift sharply from human evaluators, compressing variance, inflating surface qualities, and overlooking cultural nuance. Comparative evaluations, often assumed more robust, further establish these biases. Our analysis highlights the risks of over-relying on LLM consensus and calls for anchored meta-evaluation frameworks that integrate human disagreement, cultural diversity, and task-specific grounding.

## 1 INTRODUCTION

Evaluation has long been the engine of progress in machine learning. ImageNet (Deng et al., 2009) catalyzed the deep learning revolution in vision, while GLUE (Wang et al., 2019b) and SuperGLUE (Wang et al., 2019a) became standard bearers in language. These benchmarks worked because tasks were well-defined and success could be measured objectively. In the era of large language models (LLMs), evaluation is far less straightforward. Outputs are open-ended, correctness is subjective, and human annotation at scale is costly and inconsistent. To overcome this bottleneck, the community increasingly turns to *LLM-as-judge* frameworks (Chiang et al., 2023; Zheng et al., 2023; Wang et al., 2023), where models are used not only to generate answers but also to evaluate them.

The promise is appealing: if GPT-4 can write an essay, perhaps it can also grade one. But this raises a paradox. What happens when evaluators are themselves evaluated, or when multiple LLM judges are recursively stacked? A growing body of empirical work highlights fragilities: LLM judges exhibit strong stylistic preferences, systematically rewarding verbose and fluent answers even when they are less accurate (Wang et al., 2023; Zheng et al., 2023); they are sensitive to superficial prompt wording (Fu et al., 2024; Liu et al., 2023); they replicate their own training priors (Gilardi et al., 2023; Lin et al., 2024); and they undervalue outputs in non-English or culturally specific contexts (Xu et al., 2024; Zhuo et al., 2023). Recent surveys (Chiang et al., 2023; Fu et al., 2024; Lin et al., 2024) have documented these weaknesses, but they stop short of offering a unified account of why such brittleness is inevitable. We argue that these are not isolated artifacts but symptoms of a deeper, structural phenomenon that we call **meta-evaluation collapse**.

*Meta-evaluation* occurs whenever judgments themselves become objects of judgment: for example, when one LLM rates the evaluation provided by another, or when a human and an LLM's assessments are aggregated into a single score. This recursive structure mirrors phenomena studied in dynamical systems, cognitive psychology, and social epistemology. In psychology, metacognition, "thinking about thinking" (Flavell, 1979), can reinforce rather than correct individual biases. In social epistemology, bounded-confidence models of opinion dynamics (Hegselmann & Krause, 2002) and echo chambers (Nguyen, 2020) show how groups can converge to internally stable but externally misaligned consensus. By analogy, recursive evaluation among LLM judges risks contracting toward self-consistent equilibria that drift away from human or domain-grounded truth.

Consider two illustrative examples. First, when asked to evaluate health advice, an LLM judge may reward answers that sound fluent and empathetic while overlooking factual omissions. If such judgments are themselves re-evaluated by other LLMs, the system quickly converges on "good style" as the dominant metric, collapsing the space of evaluation to surface fluency. Second, in multilingual contexts, judges may assign near-perfect scores across languages even though humans detect clear weaknesses in lower-resource languages. Such flattening masks the very cultural and linguistic nuances that matter most in practice. In both cases, recursive evaluation leads to high agreement among LLM judges but systematic divergence from human evaluators.

Crucially, this collapse is not accidental; it is *structurally inevitable* without external anchoring. From the perspective of dynamical systems, evaluation operators act as contractions: they reduce variance and enforce agreement, but in doing so they risk locking into biased attractors. From the perspective of learning theory, it parallels the bias–variance tradeoff (Geman et al., 1992): contraction reduces variance across evaluators but amplifies systematic biases. And from the perspective of social theory, it echoes how epistemic bubbles amplify internal agreement while silencing external correction (Nguyen, 2020).

**Our contribution is to move beyond cataloguing weaknesses and to provide the first structural account of why collapse arises, formalizing meta-evaluation as a recursive operator with predictable failure modes.** This framing matters because evaluation is not a peripheral task but the driver of model progress: if evaluators collapse, so too does the trajectory of the field. We show both theoretically and empirically that recursive, unanchored LLM-based evaluation converges to internally coherent but epistemically inadequate fixed points, and we chart concrete safeguards, anchoring, pluralistic aggregation, and hybrid human–AI pipelines, to resist collapse. In short, we argue that confronting meta-evaluation collapse is necessary for building trustworthy evaluation ecosystems in the LLM era.

## 2 BACKGROUND

### 2.1 LLM-AS-JUDGE IN EVALUATION PARADIGMS

The growing cost, inconsistency, and limited scalability of human evaluation in open-ended tasks has accelerated the adoption of *LLM-as-judge* frameworks, where large language models themselves act as evaluators. This paradigm has been widely applied across dialogue and chatbot benchmarking (Zheng et al., 2023), summarization (Krishna & et al., 2023), machine translation (Kocmi & Federmann, 2023), argumentation and political discourse (Gilardi et al., 2023), and even highly specialized domains such as biomedical and legal reasoning (He et al., 2023; Bang & et al., 2023). These studies consistently report that LLM judges correlate better with human preferences than traditional automatic metrics, offering a scalable and flexible alternative to n-gram–based or embedding-based evaluation. Benchmarks such as MT-Bench, Chatbot Arena, and GPT-Eval have popularized the approach, and LLM-based evaluators are now integral to model development and deployment pipelines.

However, this apparent success masks important fragilities. Unlike humans, who bring grounded expertise and socio-cultural experience to evaluation, LLM judges are themselves trained black boxes whose scoring mechanisms are entangled with the same data, priors, and biases that shape their generative behavior. For example, GPT-4 can assign nuanced helpfulness scores, but those scores often reflect stylistic similarity to its own training distribution rather than domain-grounded correctness. The opacity of the scoring process makes it difficult to know whether agreement between models reflects genuine quality alignment or merely shared biases. In practice, evaluations often converge toward surface-level criteria such as fluency, politeness, and verbosity, which are easy to detect but not necessarily diagnostic of factual accuracy or completeness (Wang et al., 2023; Lin et al., 2024). This raises fundamental questions about the *trustworthiness* of LLM-based evaluation.

### 2.2 SYSTEMATIC WEAKNESSES OF LLM JUDGES

A growing body of work has begun to document these weaknesses in detail. One consistent finding is the tendency of LLM judges to reward fluency and verbosity over precision or factual adequacy (Wang et al., 2023; Zheng et al., 2023). Models often rate long, stylistically polished answers as superior to concise but correct ones, thereby introducing systematic inflation into evaluation pipelines.

Another recurrent issue is sensitivity to prompt framing: seemingly minor changes in rubric wording or context can lead to preference reversals, undermining replicability (Fu et al., 2024; Liu et al., 2023). Even the provision of reasoning or chain-of-thought rationales does not guarantee reliability, since these rationales can themselves be biased, post hoc justifications (Turpin et al., 2023; Wang et al., 2024), reflecting the same stylistic preferences that drive the judgments.

Perhaps more concerning is the phenomenon of self-bias and circularity. When models are tasked with evaluating their own outputs or those of closely related systems, they tend to show a preference for stylistic and distributional features aligned with their own generations (Gilardi et al., 2023). This creates a feedback loop where models reward themselves, reinforcing rather than correcting generation biases. Moreover, in multilingual and culturally diverse settings, LLM judges have been shown to penalize non-English outputs or culturally grounded expressions, resulting in evaluations that are neither representative nor fair (Xu et al., 2024; Zhuo et al., 2023). These limitations collectively suggest that consistency among LLM judges cannot be equated with validity relative to human expectations or domain-grounded truth.

## 2.3 FROM RELIABILITY TO COLLAPSE

The key tension, then, is that while LLM judges are attractive for their scalability and internal consistency, they may converge to self-consistent but externally invalid states. High inter-model agreement has been documented in multiple studies (Zheng et al., 2023; Lin et al., 2024), yet this agreement often diverges from human annotators. In scientific domains, He et al. (2023) show that LLM judges reward plausibility over rigor, while Fu et al. (2024) demonstrate that hierarchies of judges can magnify prompt artifacts rather than cancel them out. In practice, this means that recursive or aggregated evaluation pipelines risk amplifying the very biases that they are intended to smooth out.

**Analogous risks are well documented in other evaluation ecosystems.** In peer review, for example, recursive judgment and reputation effects have been shown to produce convergence toward conservative consensus, suppressing novel contributions (Lee et al., 2013; Teplitskiy et al., 2018). In algorithmic auditing, evaluation pipelines can themselves reinforce systemic bias, especially when benchmarks reflect skewed or incomplete ground truths (Raji et al., 2020; Mitchell et al., 2021). Even in scientific metrics such as citation-based impact factors, recursive reliance on aggregate scores creates distorted incentives and misaligned indicators of quality (Biagioli & Lippman, 2016). These parallels underscore that meta-evaluation collapse is not unique to LLMs but a recurring structural risk when evaluators are recursively judged without external anchoring.

This dynamic can be understood as a form of **meta-evaluation collapse**, where evaluation hierarchies built from LLM judges converge to fixed points that are internally stable but detached from external oracles such as human expertise or ground truth. Unlike isolated evaluation errors, collapse is structural: it arises from the operator-like nature of recursive judgment, where contractions reduce variance and increase inter-judge agreement, but in doing so lock the system into biased attractors. This parallels classical observations in cognitive psychology, where recursive self-reflection can entrench rather than correct biases (Flavell, 1979), and in social epistemology, where bounded-confidence dynamics yield echo chambers that are internally consistent but externally unreliable (Hegselmann & Krause, 2002; Nguyen, 2020). Recognizing these risks is essential, because evaluation is not just a secondary task but a driver of model development: if the evaluators collapse, so too will the trajectory of model progress.

## 3 THEORETICAL FOUNDATIONS OF META-EVALUATION COLLAPSE

We formalize *meta-evaluation* as a recursive process in which evaluators are themselves judged by other evaluators. This recursive framing allows us to explain why hierarchies of LLM-based judges often converge internally to stable patterns of agreement, yet fail to align with external oracles such as human judgment or ground-truth correctness. To capture these dynamics, we draw on three complementary theoretical lenses: contraction mappings from dynamical systems, spectral analysis from linear algebra, and limits from learning theory and social epistemology. Each of these theories provides tools to understand different facets of evaluation collapse: why convergence occurs, how biases are amplified, and why no evaluator can be universally aligned without external anchoring.

**A toy illustration.** Suppose two LLM judges must evaluate short answers on two criteria: fluency and factual accuracy. If Judge A slightly favors fluency and Judge B slightly favors accuracy, recursive meta-evaluation may repeatedly "average" their scores. Over time, the contraction toward consensus suppresses variance, and the system converges to a single fixed preference. In practice, this fixed point often aligns with surface fluency, because it is easier to detect consistently than factual correctness. This toy example illustrates how recursive judgment eliminates diversity of criteria and collapses evaluation space into a biased attractor.

**Evaluation as an operator.** We represent evaluation as a mapping between input prompts $\mathcal{X}$, model responses $\mathcal{Y}$, and judgments $\Omega$ (e.g., ordinal scores or probabilities). An evaluator is a conditional distribution

$$e : \mathcal{X} \times \mathcal{Y} \to \Delta(\Omega),$$

and meta-evaluation arises when we define an operator $\mathcal{T}$ that maps evaluators to evaluators: $e_{k+1} = \mathcal{T}(e_k)$. Iterating this process yields a hierarchy $\{e_0, e_1, e_2, \dots\}$, which can be studied analogously to recursive dynamical systems. This abstraction captures real practices such as "LLMs judging LLMs" or aggregating multiple evaluators into higher-order judgments.

**Contraction and convergence.** Classical results from fixed-point theory show that if $\mathcal{T}$ is a contraction with Lipschitz constant $\beta < 1$, then all sequences $\{e_k\}$ converge to a unique fixed point $e^*$ (Banach, 1922). Intuitively, this explains why LLM judges often display strong agreement: their evaluation maps act as contractions that suppress variation across items and evaluators, forcing judgments into a narrow region of the score space. However, convergence is agnostic to truth. The fixed point $e^*$ need not coincide with the oracle $O$ (e.g., human ground-truth judgment). Thus, convergence itself is not a guarantee of reliability but rather a structural property of recursive evaluation. *This theoretical contraction directly parallels the empirical ceiling effects we later observe, where LLM judges compress distributions into artificially high but homogeneous ratings.*

**Spectral view of bias.** To analyze how specific biases evolve under recursion, we linearize $\mathcal{T}$ near an evaluator $e$, yielding the recurrence $v_{k+1} = Av_k$, where $A$ is the Jacobian of $\mathcal{T}$ and $v_k$ encodes evaluation statistics. The spectral radius $\rho(A)$ governs the long-run dynamics (Meyer, 2000): if $\rho(A) < 1$, variation shrinks and judgments collapse into trivial consensus; if $\rho(A) = 1$, distortions persist without correction; if $\rho(A) > 1$, certain biases are amplified with each iteration. Eigen-directions of $A$ correspond to systematic tendencies in evaluation, for instance, a preference for fluency or verbosity, that can either contract to insensitivity or expand into runaway bias. *This spectral view provides the mathematical analogue to our empirical winrate asymmetries, where stylistic preferences (e.g., verbosity) are amplified across rounds of comparative judgment.*

**Learning-theoretic limits.** Even if biases could be controlled, learning theory warns us of deeper limitations. The No Free Lunch theorem (Wolpert, 1996) states that no algorithm performs optimally across all tasks. By analogy, no meta-evaluation operator $\mathcal{T}$ can be universally aligned with an oracle $O$ across all domains. In practice, recursive LLM judging risks forming closed evaluative systems that are internally coherent but externally misaligned. This resembles echo chambers in social epistemology (Nguyen, 2020) and bounded-confidence opinion dynamics (Hegselmann & Krause, 2002), where repeated reinforcement produces consensus detached from reality. These parallels highlight that evaluation collapse is not merely a technical bug, but a structural consequence of recursive, unanchored judgment.

**Anchoring as a safeguard.** To prevent collapse, evaluation operators must be coupled to an external oracle. Formally, we define an anchored operator

$$\mathcal{T}_\alpha(e) = (1 - \alpha)\mathcal{T}(e) + \alpha O,$$

where $\alpha \in (0, 1]$ determines the strength of anchoring to the oracle $O$. The linearization of this operator has spectral radius $(1 - \alpha)\rho(A)$, which ensures convergence to an $O$-anchored fixed point whenever $(1 - \alpha)\rho(A) < 1$. Even a small degree of anchoring suffices to stabilize the hierarchy, ensuring that evaluators converge toward external truth rather than internal bias. This insight motivates hybrid approaches where human supervision or factual ground-truth signals are periodically integrated to constrain LLM-based evaluation (Kremer et al., 2018; Gao & et al., 2023).

Taken together, these theoretical tools allow us to characterize *meta-evaluation collapse*. Contraction explains why LLM judges often appear consistent (mirroring ceiling effects in absolute ratings); spectral analysis explains how specific biases grow or vanish (mirroring bias amplification in pairwise winrates); learning-theoretic limits explain why universal correctness is impossible without oracles; and anchoring offers a way to stabilize evaluation against collapse. Crucially, collapse is not a contingent failure mode but a structural inevitability of recursive, unanchored evaluation hierarchies, a risk that becomes especially salient in high-stakes, culturally nuanced domains.

## 4 EXPERIMENTAL SETUP

Our theoretical framework predicts that recursive LLM-based judging contracts evaluations into internally consistent but externally misaligned fixed points. To rigorously test this in practice, we moved beyond synthetic benchmarks and designed a community-centered setup in the multilingual health domain, where judgments are socially consequential, culturally nuanced, and deeply entangled with local practices. This choice was motivated by prior critiques of benchmark-driven evaluation that highlight the risks of decontextualized tasks (Paullada et al., 2021; Sambasivan et al., 2021). By grounding evaluation in real-world health dilemmas, we aimed to capture the precise conditions where meta-evaluation collapse becomes most consequential.

**Community-centered data collection.** Data was curated in partnership with Civil Society Organizations (CSOs) and local data workers who have lived experience navigating health systems in India. Unlike typical evaluation datasets, these queries were not designed to be synthetic or templated. Instead, they were drawn from practical dilemmas faced in everyday contexts: questions balancing modern medical practices with traditional remedies, navigating intergenerational disagreements over treatment, or understanding conflicting advice from formal and informal sources. This participatory and culturally embedded approach ensured that the evaluation space reflected not only linguistic diversity but also the pragmatic and social complexities that real users confront. The resulting dataset comprises 270 queries in each of three languages, Hindi, Kannada, and Malayalam, yielding 810 community-driven test cases. Each query is multi-layered, embedding factual uncertainty alongside cultural and social context. Importantly, this dataset was curated outside of model training corpora, creating what we call a *community-driven benchmark* for health.

**LLM answer generation.** Using this dataset, we generated model responses with three state-of-the-art multilingual LLMs: Sarvam-M (AI, 2024b), Qwen3-235B-A22B (Team, 2024), and Llama-3.1-405B-Instruct (AI, 2024a). Each query was prompted using a consistent template specifying the role of the model as a "health expert," constraining output language (Hindi, Kannada, or Malayalam), and fixing word limits to mitigate verbosity biases. Decoding parameters followed the official defaults recommended by each model, ensuring comparability without introducing artificial discrepancies. This pipeline produced three responses per query, yielding 2,430 total responses (810 queries × 3 models).

**Evaluation design.** Evaluation criteria were co-developed with CSO partners to reflect what real users value when engaging with health information. The four dimensions were: (i) clarity and fluency, assessing whether the answer is linguistically accessible; (ii) helpfulness and relevance, examining whether the answer addresses the user's actual concern; (iii) accuracy, capturing factual and epistemic reliability; and (iv) completeness versus conciseness, assessing whether sufficient information is provided without redundancy. These dimensions go beyond surface-level linguistic quality to embed socio-pragmatic expectations, which theory predicts LLM judges will systematically compress or overlook. Both standalone and comparative paradigms were employed: standalone ratings yield absolute distributions to probe ceiling effects and contraction, while comparative pairwise judgments reveal subtle discriminations and expose bias amplification (Agarwal et al., 2020; Dong et al., 2020).

**LLM-as-judge evaluation.** Two LLMs served as evaluators. GPT-4o (OpenAI, 2024) represented a closed-source, high-performing baseline widely used in evaluation studies. Sarvam-M doubled as both evaluated system and evaluator, creating conditions to test for self-preference and circularity, a form of recursive self-reinforcement that theory predicts may accelerate collapse. Both judges

provided categorical judgments and free-text rationales for each evaluation, making their scoring decisions interpretable and analyzable in light of collapse dynamics.

**Human evaluation.** To provide a cultural and epistemic anchor, we recruited 23 human evaluators who were native speakers of the three languages. Twelve of these evaluators also participated in the query creation phase, ensuring continuity of cultural nuance across data generation and evaluation. Human judges followed the same rubric as LLMs but supplemented scores with both written explanations and spoken voice notes detailing their reasoning process. This procedure allowed us to capture not only judgments but also the underlying cultural and pragmatic rationales often invisible in surface-level scores. Human evaluation thus plays a dual role: as an external oracle against which LLM collapse can be measured, and as an interpretive lens into what dimensions of judgment collapse discards.

**Evaluation strategy.** The final evaluation corpus contained all judgments, human and LLM, linked to each query, response, and criterion. Standalone judgments captured distributional compression, while comparative judgments operationalized preference blocs and divergence. To ensure robustness, we employed consistent tie-handling rules in pairwise aggregation, used bootstrap resampling for uncertainty estimates, and applied multiple-hypothesis correction where appropriate. This comprehensive design connects directly with our theoretical framework: contraction and ceiling effects appear in standalone ratings, bias amplification emerges in pairwise comparisons, and divergence between LLMs and humans exposes the impossibility of unanchored convergence.

## 5 EMPIRICAL ANALYSIS

We connect the operator-theoretic account of evaluation collapse with empirical evidence from both standalone and comparative judgments. Across all settings, LLM judges converge to compressed, internally consistent evaluations that drift systematically from human anchors.

### 5.1 COMPRESSION IN STANDALONE RATINGS

Absolute scores show clear ceiling effects: LLM judges assign higher, tightly clustered scores, while humans produce lower and more dispersed ratings. This contraction is visible when disaggregated by criterion and language (see Figure 1). Automated judges inflate clarity and helpfulness, while humans penalize completeness more severely; and they show uniform scores across Hindi, Kannada, and Malayalam, while humans reveal stronger penalties in Malayalam. These findings illustrate the theoretical prediction that automated judges project item vectors into a narrow subspace aligned with surface fluency, obscuring culturally grounded variation.

### 5.2 AGREEMENT AND DIVERGENCE ACROSS EVALUATORS

The collapse dynamics become more explicit when examining agreement and divergence. Correlation and pairwise agreement reveal that LLM judges align strongly with each other but only weakly with humans. Variance ratios confirm that human ratings exhibit substantially more dispersion than automated judges, often by orders of magnitude. These compressed distributions explain why LLMs form a stable consensus but fail to capture the richness of human variability.

At the same time, divergence metrics show structured misalignment. Wasserstein distances highlight systematic distributional shifts between LLM and human judgments, largest for completeness and in Malayalam. Spectral decomposition identifies the dominant direction of collapse: suppression of Llama405b and alignment with Qwen3, consistent with stylistic amplification. Together, these results demonstrate that automated judges do not merely add noise but converge toward a biased evaluative attractor (see Figure 3).

### 5.3 COMPARATIVE JUDGMENTS AND BIAS AMPLIFICATION

Pairwise winrates show that humans consistently prefer Qwen3, while LLM judges favor Sarvamm, particularly in Hindi and Malayalam (see Figure 4). This asymmetry illustrates bias amplification: automated judges reward stylistic similarity to their own outputs, while humans penalize missing

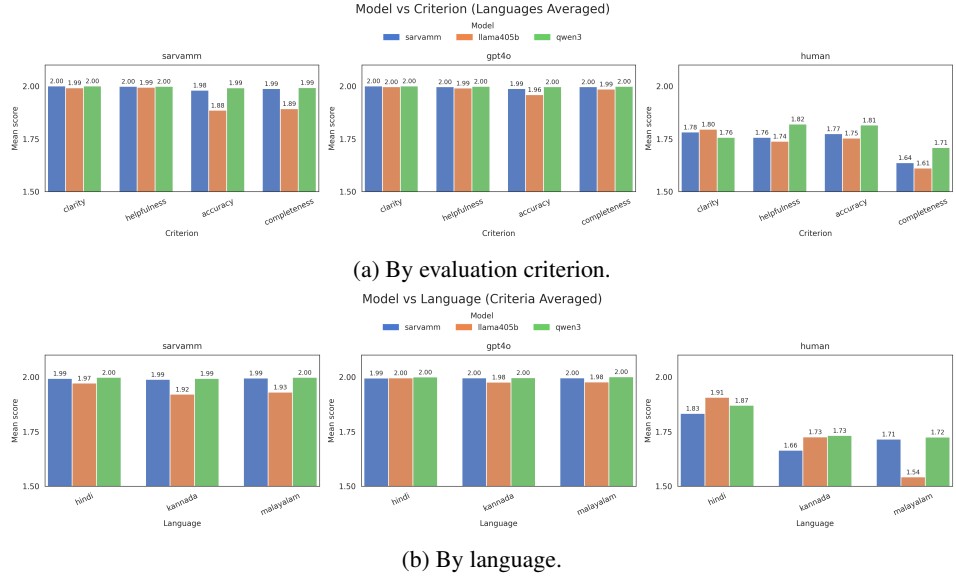

(a) By evaluation criterion.

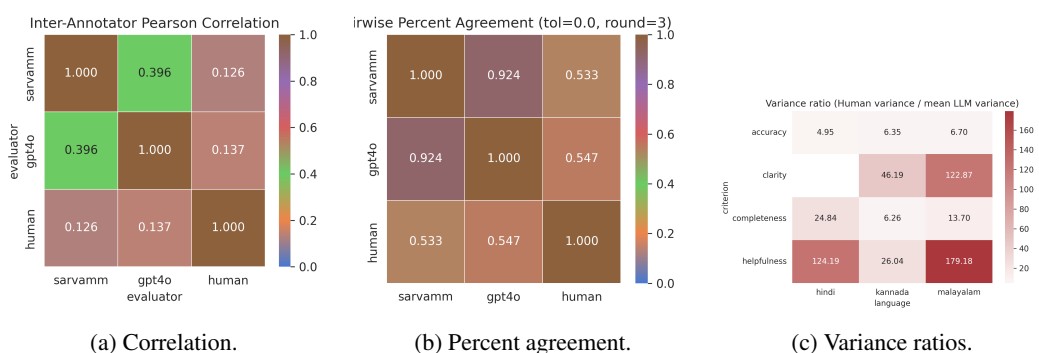

(b) By language.

Figure 1: Standalone absolute ratings. LLM judges compress scores near the ceiling, while humans assign lower, more varied ratings.

(a) Correlation.

(b) Percent agreement.

(c) Variance ratios.

Figure 2: Agreement and dispersion across evaluators. Automated judges converge tightly with one another but fail to capture the richer variance observed in human ratings.

or incomplete information. Comparative evaluation, often treated as more robust, thus inherits the same collapse dynamics.

## 5.4 OPERATOR DYNAMICS AND GLOBAL COLLAPSE

Finally, operator-level diagnostics quantify the global structure of collapse. Spectral radii show LLM-to-LLM mappings hover near unity, reinforcing consensus without recovering nuance, while LLM-to-human mappings expand strongly and human-to-LLM mappings contract sharply. Kappa statistics confirm moderate inter-LLM agreement but weak LLM–human alignment. Entropy analysis shows that automated judges exhibit confident, low-entropy preferences, while humans remain uncertain on borderline cases. Together, these operator dynamics reveal a low-dimensional attractor: automated judges compress variance, amplify stylistic bias, and drift systematically from the richer human evaluation space (see Figure 5).

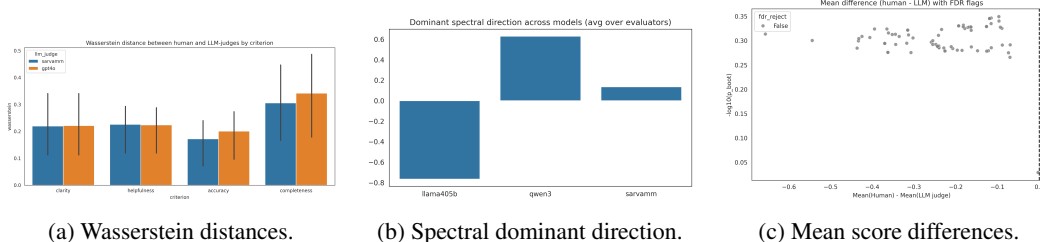

(a) Wasserstein distances.   (b) Spectral dominant direction.   (c) Mean score differences.

Figure 3: Divergence diagnostics. Human judgments diverge systematically from LLMs, with strongest gaps on completeness and Malayalam, aligned with the dominant spectral direction of collapse.

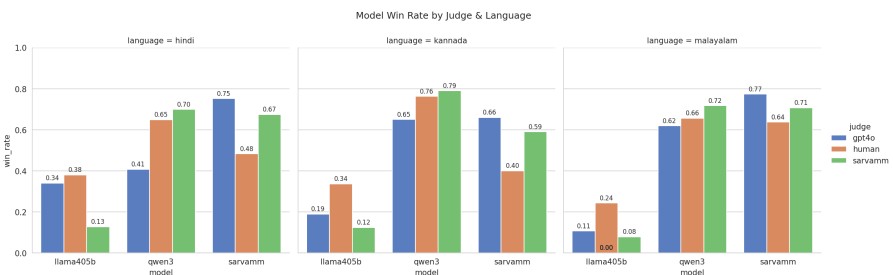

Figure 4: Comparative winrates by language. Humans prefer Qwen3, while LLM judges favor Sarvamm, reflecting stylistic amplification.

## 6 DISCUSSION

**Findings of collapse.** Our findings reveal a fundamental tension in the use of LLMs as evaluators. Across both standalone and comparative settings, LLM judges displayed high internal consistency, compressing scores into narrow distributions and aligning strongly with one another. Yet this very consistency masks a deeper misalignment: relative to human evaluators, LLM judgments systematically inflate surface qualities such as fluency and clarity while underweighting integrative dimensions such as completeness and cultural adequacy. The result is an epistemically coherent but externally biased evaluation landscape, a phenomenon we formalize as meta-evaluation collapse.

**Contraction, amplification, and echo chambers.** Several aspects of this collapse deserve emphasis. First, contraction effects dominate: variance ratio analyses showed that LLM judges suppress the dispersion of scores by one to two orders of magnitude relative to humans, particularly on socially consequential criteria like helpfulness and completeness. This variance compression produces the illusion of consensus but erases the nuance that human evaluators bring to culturally embedded queries. Second, amplification effects emerge in comparative settings. Entropy diagnostics demonstrated that LLMs frequently resolve borderline cases with high confidence, where humans exhibit genuine disagreement. This suggests that recursive LLM-based pipelines do not merely contract judgments but can also entrench systematic preferences in eigen-directions of bias, rewarding fluency, penalizing brevity, or favoring outputs aligned with their own generative style. Third, echo-chamber dynamics are visible in inter-evaluator agreement. While Sarvam-M and GPT-4o correlate at near-fixed-point levels, their alignment with humans remains weak, quantifying the drift from oracles that our theoretical model predicts.

**Design principles for future evaluation ecosystems.** Our results motivate several design principles for building more robust evaluation systems:

- **Anchoring through hybrid pipelines.** Periodic calibration against human raters or ground-truth oracles prevents collapse into self-reinforcing equilibria (Kremer et al., 2018; Gao & et al., 2023).

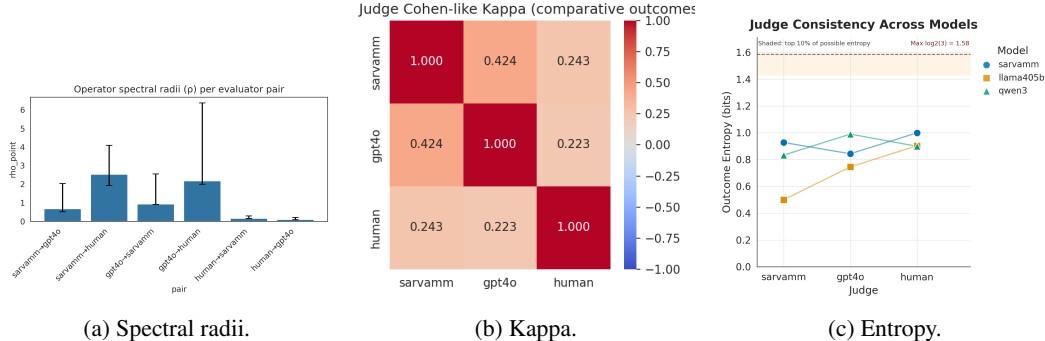

(a) Spectral radii.        (b) Kappa.        (c) Entropy.

Figure 5: Operator dynamics of collapse. Automated judges reinforce each other, compress human variance, and exhibit low-entropy preferences detached from human anchors.

- **Pluralistic aggregation.** Combining multiple evaluators, LLMs and humans, reduces the risk of bias dominance and captures disagreement, echoing ensemble methods and pluralistic annotation practices (Snow et al., 2008; Zheng et al., 2023; Lin et al., 2024).

- **Entropy-aware routing.** Prior work in uncertainty-aware LLM evaluation suggests using confidence or entropy to decide when to defer to humans (Nguyen & Chen, 2023; Kadavath et al., 2022). Such routing preserves scalability while safeguarding epistemic fidelity.

- **Bias-corrective operators.** Spectral diagnostics and re-weighting can explicitly suppress eigen-directions associated with stylistic amplification, drawing on ideas from fairness calibration and debiasing in evaluation (Wang et al., 2023; Liu et al., 2023; Mitchell et al., 2023).

Together, these principles move beyond ad hoc fixes toward systematic design for resilience against collapse.

**Open research challenges.** Several open challenges emerge from this work. Can we design evaluators that not only approximate human judgments, but also model the *distribution of human disagreement*, thereby capturing epistemic uncertainty as a first-class signal (Davani et al., 2022; Pavlick & Kwiatkowski, 2022)? How can evaluation protocols embed culturally situated ground truths, especially in multilingual and health-related contexts where the risks of collapse are highest (Paullada et al., 2021; Sambasivan et al., 2021)? What weighting schemes best combine heterogeneous evaluators into meta-operators that resist collapse, rather than reinforce it? More provocatively: is it possible to build *self-aware evaluators* that can detect when they are collapsing into biased attractors and flag the need for external anchoring?

**Toward robust evaluation ecosystems.** Ultimately, our results argue for a paradigm shift. The promise of LLM-as-judge lies not in supplanting human evaluation but in scaling it responsibly through hybrid pipelines that recognize and correct structural biases. Anchoring evaluators, embracing pluralistic judgment, and designing diagnostics for collapse are essential steps toward trustworthy evaluation ecosystems. Without such safeguards, recursive use of LLM judges risks reinforcing false consensus, obscuring real cultural variance, and eroding the reliability of evaluation benchmarks themselves. By exposing the inevitability of collapse and charting concrete paths forward, this work aims to establish the foundations for a new generation of evaluation methods that are scalable, interpretable, and epistemically robust.

## 7 CONCLUSION

LLM judges promise scalability but collapse into self-consistent yet misaligned evaluations, compressing nuance and amplifying bias. Our analysis shows that inter-model agreement is not a measure of truth but a structural drift away from human judgment. Building resilient evaluation requires anchoring to human disagreement, embracing cultural diversity, and designing pluralistic frameworks that resist collapse and safeguard the trajectory of progress.

# 8 REPRODUCIBILITY STATEMENT

To ensure reproducibility of our LLM and human evaluation and subsequent analysis, we provide both data samples and code in the supplementary material. The submission includes a `data` directory and a `code` directory.

The `data` directory contains representative samples of the inputs and evaluation records: `sample_User_Questions.csv`, which provides a sample of user queries collected through the community-centered process, and `Sample_LLM_as_judge_and_Human_Evaluation_Score.csv`, which includes example records of both LLM-judge explanations and human evaluation scores for standalone and comparative judgments.

The `code` directory contains Python scripts and configuration files for reproducing the main analyses, including both standalone and comparative evaluation pipelines. All scripts are documented with configuration files to enable exact replication of our results.

We will publicly release the **code** associated with this work following publication.

# 9 ETHICAL CONSIDERATIONS

Our ethical considerations follow the framework proposed by Bender & Friedman (2018), with attention to institutional oversight, data provenance, and annotator welfare.

**Institutional Process and Oversight**   The community-centered data collection was conducted in collaboration with Civil Society Organizations (CSOs) and local data workers. All participants were briefed about the purpose of the study, and consent was obtained prior to participation.[1]

**Data Provenance and Quality Assurance**   The dataset was created through participatory design with CSOs to reflect real-world, multilingual health queries. This ensured that the evaluation captured culturally grounded and socially consequential dilemmas. To minimize potential harms and ensure quality, all responses were reviewed by both human evaluators and automated tools. No sensitive personal information was included in the dataset.

**Annotator Demographics**   Fifteen community data workers contributed to the creation of queries, representing diverse age, gender, and educational backgrounds (Table 1). Their linguistic expertise and lived experience were crucial for ensuring that the data reflected genuine community priorities. The CSOs that facilitated collaboration and deployment contexts are summarized in Table 2.

| | |
|---|---|
| **Number of participants** | 15 |
| **Age range** | 19–36 |
| **Gender** | Male: 4 ; Female: 11 |
| **Education** | High-school: 1 ; Undergraduate: 10 ; Graduate: 4 |

Table 1: Demographic details of community data workers.

| CSO ID | Chatbot deployment | Languages supported |
|---|---|---|
| CSO-1 | Yes | Hindi, Marathi, Telugu |
| CSO-2 | Yes | Kannada, English |
| CSO-3 | Yes | Hindi, English |
| CSO-4 | No | Malayalam |
| CSO-5 | Yes | Hindi, Kannada, English |

Table 2: Participating CSOs. Details of organizations remain anonymized for confidentiality.

---

[1]We do not disclose the names or organizational details of the CSOs or data workers to preserve anonymity.

**Use of LLMs in Research Process** We used large language models to assist in polishing the writing of this paper and for exploring related work. However, all substantive analyses, evaluations, and interpretations were performed and validated manually by the authors to ensure accuracy and accountability.

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
