# OpenReview forum: "Meta-Evaluation Collapse: Who Judges the Judges of Judges?"
_ICLR.cc/2026/Conference — ICLR 2026 Conference Withdrawn Submission_

### Official Review · Reviewer_u27H · 2025-10-24

**Soundness:** 2
**Presentation:** 3
**Contribution:** 2
**Rating:** 4
**Confidence:** 3

**Summary:**

This paper introduces and studies the phenomenon of meta-evaluation collapse. The authors argue that when large language models evaluate one another, the evaluation process converges toward internally consistent but externally biased judgements. Judgements prefer stylistic fluency and consensus over factual correctness or cultural nuance. The paper provides an empirical validation in a multilingual, community health domain, where LLM evaluators diverge from, and produce less diverse judgements than, human judges. Finally, the work suggests design principles, involving human validation, for more trustworthy LLM evaluation systems.

**Strengths:**

1.	The paper introduces the thought-provoking notion of meta-evaluation collapse, a timely concept in an era increasingly concerned with the potential erosion of diversity of thought amid AI-generated media; the operator-theoretic and dynamical-systems perspectives provide a refreshing interdisciplinary lens on what is usually treated as an empirical issue.
2.	The community-centered health dataset is thoughtfully designed and highlights the social consequences of collapse in multilingual settings. The recruitment of human annotators across three languages is a clear strength.
3.	The paper is well written and conceptually coherent; several sections (e.g., “Human evaluation thus plays a dual role…”) are insightful in articulating the epistemic stakes.
4.	The paper offers concrete suggestions (often involving humans-in-the-loop) for ways forward.

**Weaknesses:**

1.	Justification of Theoretical Tools Chosen. The use of contraction mappings and learning-theoretic limits is intriguing but under-motivated. Provide citations or examples from the original domains (e.g., dynamical systems, social epistemology) that justify why these frameworks are appropriate for analyzing loss of variation in judge LLMs. A short intuitive explanation of why they capture the phenomenon more faithfully than, say, information-theoretic or Bayesian frameworks would make the theory more persuasive.
2.	Recursion. The paper’s central argument concerns recursive evaluation, yet the experiments demonstrate only a single level of recursion. It would strengthen the work to explore deeper loops (e.g. a judge-on-judge-on-judge setup) to observe how collapse compounds over multiple iterations. Alternatively, a more practically grounded experiment could again be to examine conversational self-reinforcement, where a model engages in back-and-forth dialogue with itself: will it progressively reinforce its own beliefs?
3.	Given the emphasis on echo-chamber dynamics, it would be interesting to perform the additional (related, yet distinct) experiment in which two LLMs engage in iterative dialogue to reinforces shared biases.
4.	Previous works on multi-agent debate. The finding that recursive evaluation contracts diversity seems to run counter to prior works showing value in inter-model disagreement as a proxy for uncertainty (e.g., Du et al. 2023 on multi-agent debate). Please explicitly contrast this work with the findings in this line of work.
5.	The empirical analysis is limited to a single domain (community health). While this setting is socially significant, demonstrating meta-evaluation collapse across additional domains (e.g., open-ended dialogue) would substantially strengthen the claim that collapse is a general structural phenomenon rather than a domain-specific phenomenon.
6.	Figure 5 displays a main finding, so it would benefit the readers to define precisely how spectral radius, kappa, and outcome entropy are computed in this context and link them back to their theoretical analogues in Section 3. The paper could benefit from a short subsection or appendix explicitly mapping theoretical constructs to empirical estimators.

**Questions:**

Please see the concerns that appear in weaknesses.

---

### Official Review · Reviewer_g7hF · 2025-10-29

**Soundness:** 2
**Presentation:** 2
**Contribution:** 2
**Rating:** 2
**Confidence:** 3

**Summary:**

The paper introduces meta-evaluation collapse: recursive LLM judging converges to self-consistent but fragile fixed points detached from ground truth. Operator-theoretic analysis and multilingual health-query experiments show that unanchored hierarchies contract to biased equilibria—amplifying fluency over accuracy—so LLM judges agree with each other while diverging from humans, underscoring the need for anchored meta-evaluation.

**Strengths:**

The paper comprehensively summarizes issues in prior work on LLM judges, enabling readers to quickly grasp the significance of studying this phenomenon.

**Weaknesses:**

1. The paper draws on theories from multiple fields to support its hypothesis of meta-evaluation collapse, while the links between these theories and the proposed notion remain largely at the level of intuitive similarity, without substantive analysis.
2. The experiments are misaligned with the theoretical constructs, hindering logical validation (See questions below). Section 3 is based primarily on simple assumptions and theoretical derivations, lacking concrete, real-world cases to substantiate the arguments.
3. There are several content and formatting problems, including: (i) no in-text reference or explanation for Figure 2; (ii) incomplete text in Figure 2(b); (iii) text in several figures is too small to be legible—revision is recommended.

**Questions:**

The paper defines “meta-evaluation” as “judgments themselves become objects of judgment” (lines 96-98), i.e., using one LLM to evaluate another LLM’s evaluations. The theoretical foundation is built around this concept, defining the evaluation process (e) and the mapping between evaluations (T). However, the experiments instead ask an LLM to evaluate LLM-generated responses, which is **inconsistent** with the earlier discussion and definitions, and thus does not constitute a valid test of the proposed theory.

In addition, the experimental conclusions are broad and somewhat obvious—for example, “Comparing to human evaluators, LLMs judges tend to assign higher, tightly clustered scores” and “agreement among LLM judges is high, whereas agreement between human and LLM judges is relatively low.” These findings have already been reported in prior work and are not directly tied to the earlier theoretical claims.

---

### Official Review · Reviewer_MoNo · 2025-10-30

**Soundness:** 2
**Presentation:** 3
**Contribution:** 2
**Rating:** 2
**Confidence:** 3

**Summary:**

This paper raises the issue of metaevaluation collapse when using LLM judges. They present several theoretical lenses on the collapse of iterated evaluations. They follow this with the introduction of a new benchmark of multilingual health queries, where they compare human and LLM judges across the dataset.

**Strengths:**

The idea of experimenting with meta-evaluation is immensely valuable, and I was excited to read this paper. The evaluation set up, creating a new multilingual benchmarking dataset with human expert evaluation, is also a meaningful contribution, and generally well done.

**Weaknesses:**

The actual execution of this paper attempts too many different things and makes the overall message muddled.

The theoretical arguments about meta-evaluation collapse are drawn from many different literatures, but each one is gestured to, rather than thoroughly applied. I would recommend entirely discarding most of this section. My biggest issue is that the theoretical scenario described has no basis in actual evaluation practice. If LLMs were used to judge LLMs, the prompts used for the second round of judging would not be the same as the first, making the mathematical treatments of repeated application of the same transformations irrelevant.

For example, the "toy example" (lines 162-169) describes a scenario where LLM judges are repeatedly applied to the same text and averaged together, calling this "meta-evaluation". But this is not correct. The argument is broadly along the lines of the central limit theorem, that averaging repeated samples will reduce the variance of the estimate. But 1) repeated samples from the same LLM are not independent, and 2) they implicitly assume that the estimators are biased, since otherwise converging to the true score would be desirable!

The remaining arguments tend to deal with LLMs judging LLMs. However, these arguments are also incomplete. They use the analogy of linearizing the operation of an LLM as a function of a text, v_{k+1}=Av_{k}. This is a massive assumption that is not supported.

The experimental work in this paper is much more valuable than the theoretical work, but is not really related to the topic of meta-evaluation. I would be happy to see a full paper presenting the work with CSOs to create a multilingual benchmark for healthcare interactions in Indian languages. I would also be happy to see an analysis of cross-lingual biases in LLM judges, though I believe this has already been shown in some cases. However, I do not see an analysis of the results of repeated application of LLM judges to judge other LLM judges, which is the purported subject of the work. Figure 5 is the closest thing, showing "spectral radii" but there is no description of how these are obtained.

**Questions:**

1. Why create a new benchmark for this paper? Surely a more useful result would be to show that this issue occurs on an well-known benchmark that uses LLM judges. The new benchmark is an expensive undertaking that deserves its own paper.

2. Why do you describe inter-LLM agreement as "meta evaluation"?

3. Which of the theoretical framings best represents the way that LLM judges are most commonly used?

**Details Of Ethics Concerns:**

The paper uses human annotators but makes no mention of compensation.

---

### Official Review · Reviewer_xJ7r · 2025-10-31

**Soundness:** 2
**Presentation:** 3
**Contribution:** 2
**Rating:** 4
**Confidence:** 3

**Summary:**

- Defines meta-evaluation collapse where recursive LLM-as-judge pipelines converge to internally consistent yet externally misaligned fixed points.
- Frames evaluation as an operator with contraction and spectral perspectives, arguing that inter-LLM agreement is not validity.
- Empirically shows ceiling compression, strong inter-LLM consensus, and divergence from human distributions on multilingual health queries; pairwise setups inherit similar collapse patterns.
- Proposes safeguards such as human anchoring, pluralistic aggregation, and entropy-aware routing with practical guidance for evaluation design.

**Strengths:**

- Coherent synthesis linking operator intuition to observed ceiling effects and human divergence.
- Community-centered, multilingual setting increases ecological validity relative to synthetic benchmarks.
- Practical, testable recommendations for anchoring and pluralistic aggregation.
- Broad empirical look across standalone ratings, pairwise comparisons, and operator-level diagnostics.

**Weaknesses:**

- Limited domains and languages restrict generality; inevitability claims feel overstated.
  - Empirical scope is narrow and may conflate cultural variance with judgment quality.
- Key theoretical conditions are unmeasured on real pipelines, including Lipschitz bounds and spectra.
- Insufficient statistical rigor and missing baselines or comparisons to existing mitigation methods.
- No quantitative validation or cost analysis for the proposed safeguards.

**Questions:**

- Can you empirically bound β and the spectral radius for real judge pipelines via controlled perturbations?
- How sensitive are collapse dynamics to prompt templates and to iterative critique-refine judging?
- Does increasing evaluator diversity with larger LLM ensembles reduce collapse or merely shift the attractor?
- How do safeguards change collapse signatures on well-defined tasks versus subjective ones?

---

### Note · Authors · 2025-12-13

I have read and agree with the venue's withdrawal policy on behalf of myself and my co-authors.